# The Effects of Adding Hempseed Cake on Sperm Traits, Body Weight, Haematological and Biochemical Parameters in Rabbit Males

**DOI:** 10.3390/vetsci11100509

**Published:** 2024-10-16

**Authors:** Andrej Baláži, Andrea Svoradová, Anton Kováčik, Jaromír Vašíček, Peter Chrenek

**Affiliations:** 1Institute of Farm Animal Genetics and Reproduction, NPPC, Research Institute for Animal Production in Nitra, Hlohovecká 2, 951 41 Lužianky, Slovakia; andrea.svoradova@nppc.sk (A.S.); jaromir.vasicek@nppc.sk (J.V.); 2Faculty of AgriSciences, Mendel University in Brno, 613 00 Brno, Czech Republic; 3Faculty of Biotechnology and Food Science, Slovak University of Agriculture in Nitra, 949 76 Nitra, Slovakia; anton.kovacik@uniag.sk

**Keywords:** hempseed cake, *Oryctolagus cuniculus*, nutrition, performance, semen quality, health status

## Abstract

The integration of agro-industrial residues into animals’ diet offers a major potential for advancing of a circular economy, enhancing both economic and environmental sustainability. Hempseed cake is widely utilized around the globe as a source of food and supplement ingredients. Hempseed cake has a pleasant nutty flavour and is a valuable source of essential fatty acids, minerals, vitamins, and fibres. It also contains essential amino acids within its highly digestible proteins. Feeding with a hempseed cake in rabbits, given at both tested concentrations, had no effect on weight gain per week and the total average weight gain compared to the control group. Hempseed cake addition did not decrease sperm concentration in ejaculate, sperm motility, and progressive motility. Selected haematological and biochemical indexes were examined. No negative effects of hempseed cake feeding on male rabbit reproduction and health status were found.

## 1. Introduction

Agro-industrial co-products can decrease the feeding cost and improve animal products in terms of quality and sustainability. In this context, agro-industrial co-products or former foods could serve as valuable feed ingredients in animal nutrition, helping to reduce the environmental impact of waste. The co-products are derived from various agro-industrial processes, such as the production of oil, sugar, fruit juice, and canned or frozen vegetables, as well as from roots and tubers [1]. Some co-products such as beet pulp, corn gluten feed, soybean (hulls, meal, and molasses), and sunflower meal are largely used as animal feedstuffs as sources of fibre, protein, and sugar [2,3,4]. However, new agro-industrial crops such as cardoon (*Cynara cardunculus* L.) and hemp (*Cannabis sativa* L.) are emerging within the last few years [3,5]. Hempseed typically contains over about 30% oil, 25% protein, 34% carbohydrates, dietary fibre, vitamins, and minerals [6]. *Cannabis sativa* cultivars grown for industrial purposes are characterized by lower levels of delta-9-tetrahydrocannabinoid (THC), the active principle responsible for cannabis’s psychotropic effects [7]. Hemp is an extraordinary crop, boasting immense social and economic value. Furthermore, hempseed oil serves as a source of healthy polyunsaturated fatty acids (PUFAs), and hemp sprouts are rich in antioxidants [8,9,10]. It is worth noting that hempseed oil can contain trace amounts of THC and the other cannabinoids, which are lipophilic antioxidants having therapeutic potential [6]. However, due to the high polyunsaturated fatty acid composition of hemp seed, the oil is highly susceptible to oxidative deterioration during long-term storage or preparing food by heat processing [11]. After extracting the oil, the remaining hempseed cake can be used as a protein feed for rabbits, for example.

Rabbit meat production is especially interesting because rabbits can easily convert the available proteins from cellulose-rich plants, i.e., efficiently convert fodder to food [12]. The gastrointestinal physiology of rabbits is a complex system that revolves around the separation of digestible and indigestible components of the diet in the proximal colon. The rabbit has a digestive system which allows for the reingestion and absorption of bacteria and their by-products in the small intestine [13]. The data on rabbit semen composition, including content in various components and minerals, as well as the male rabbit reproductive system and litter characteristics in relation to parity order are reviewed in previous papers [14,15,16]. In contrast to other domestic species, semen from rabbits is characterized by its low volume and sperm concentration (0.4–0.5 mL and 150–500 × 10^6^ sperm/mL, respectively) [14,17]. These characteristics also vary depending on the breed or genetic line [17,18]. Rabbit semen contains numerous droplets and vesicles, which may have distinct roles in modulating motility, capacitation, and the acrosome reaction [18].

So far, no data from fattening with hempseed cake experiments in rabbits have been published until now. The aim of our study was to investigate the effect of dietary use of hempseed cake in the rabbit’s diet on growth performances, sperm quality, haematology, and serum biochemistry parameters. Potential use of agro-industrial co-products in the animal diet could decrease the feeding cost. In particular, it is expected that feeding of hempseed cake could bring improvement in several performance traits like body weight gains and reproductive health parameters.

## 2. Materials and Methods

### 2.1. Ethical Statement

The treatment of the animals was approved by the Ministry of Agriculture and Rural Development of the Slovak Republic under the reference numbers SK U 18016 and SK P 28004. The procedures followed ethical guidelines presented in Slovak Animal Protection Regulation, RD 377/2012, which conforms to European Union Regulation 2010/63.

### 2.2. Animals

Clinically healthy male rabbits of the New Zealand White line from the National Agricultural and Food Centre—RIAP Nitra (Slovak Republic) were utilized in this experiment. The animals (n = 30; 5 months old) were housed in individual metal enriched cages (dimensions L = 78 cm, W = 58 cm, H = 50 cm), under a constant photoperiod of 16 h light and 8 h dark throughout the experiment. Temperature and humidity in the building were recorded continuously by means of a thermograph positioned at the same level as the cages (average relative humidity and temperature during the year was 60 ± 5% and 17 ± 3 °C). The rabbits in the control group (C; n = 10) were fed ad libitum with a commercially available complete feed mixture without antibiotics (KV; SIGI TRADE, Ltd., Dvory nad Žitavou, Slovak Republic) comprising pellets 3.5 mm in diameter. The rabbits in the first experimental group (E5; n = 10) were fed granulated mixture including 5% hempseed cake, and in the second experimental group (E10; n = 10) were fed granulated mixture including 10% hempseed cake, equally without addition of antibiotics. Hempseed cake (Figure 1) was produced at the Food Research Institute of the National Agricultural and Food Centre (Bratislava, Slovak Republic). The diet formulation for all groups is presented in Table 1. The males were fed with standard food (C) or food enriched with hempseed cake (E5, E10, Table 2) during 4 months and they were weighed weekly. Although the animals were already adults, average weekly and total weight gain were recorded, and their health status was also monitored. Clean tap water was provided using nipple drinkers ad libitum. The samples of individual feeds were analysed for content of nutrients (Table 1 and Table 2), according to the procedures of the AOAC (2005) [19], and starch, according to the alpha-amyloglucosidase method. The content of metabolizable energy of sample mixture was calculated by the equation of Wiseman et al. (1992) [20].

### 2.3. Semen Collection and Handling

Semen was collected from sexually mature male rabbits using a pre-heated artificial vagina two times a week. The semen was taken to the laboratory in a water bath at 37 °C and processed individually for an in vitro evaluation as described earlier [21].

### 2.4. Measurement of Spermatozoa Characteristics Using Flow Cytometry

For flow cytometry, sperm samples (aliquots from each rabbit semen) were diluted to the concentration of 1 × 10^6^ sperm in a phosphate buffer saline (PBS; Ca^−^ and Mg^−^ free, Life Technologies, Bratislava, Slovak Republic) before incubation with a specific fluorescent dye. Each sample was stained with a fluorescent dye specific for different sperm characteristics, e.g., viability, apoptosis-like changes, acrosome integrity, etc., was co-stained with DRAQ5 staining solution (100 μmol·mL^−1^; BioLegend, San Diego, CA, USA) to distinguish nucleated cells from seminal granules. The viability of spermatozoa was analysed using SYBR-14, a membrane-permeant nucleic acid green-fluorescent dye (LIVE/DEAD**^®^** Sperm Viability Kit; Thermo Fisher Scientific, Waltham, MA, USA, at final concentration of 100 nM) incubated with sperm in the dark at 37 °C for 10 min. The proportion of dead cells was determined by SYTOX Green staining solution (30 µmol·mL^−1^; Thermo Fisher Scientific, Waltham, MA, USA) incubated with sperm in the dark at 37 °C for 15 min.

The proportion of apoptotic-like cells was evaluated using YO-PRO-1 (100 µM; Molecular Probes, Lucerne, Switzerland, incubated at RT in the dark for 15 min) and Caspase 3/7 (CellEvent™ Caspase-3/7 Green Flow Cytometry Assay Kit; Thermo Fisher Scientific, Waltham, MA, USA, incubated at 37 °C in the dark for 30 min), which specifically recognizes active caspase-3 and caspase-7 proteins.

Further, the evaluation of acrosome integrity was performed using a fluorescein-labelled lectin from peanut agglutinin (*Arachis hypogea*; PNA Alexa Fluor 488, Molecular Probes, Lucerne, Switzerland, incubated at RT in the dark for 15 min).

The activity of mitochondria was evaluated using MitoTracker**^®^** Green FM (MT Green; Thermo Fisher Scientific, Waltham, MA, USA, incubated at 37 °C in the dark for 10 min).

Capacitation of rabbit spermatozoa was evaluated via specific Ca^2+^ green-fluorescent probe FLUO-4 AM (FLUO-4; Thermo Fisher Scientific, Waltham, MA, USA, incubated at RT in the dark for 15 min).

In addition, reactive oxygen species (ROS) production was assessed by CellROX dye (2.5 μM, Thermo Fisher Scientific, Waltham, MA, USA, incubated at 37 °C in the dark for 15 min). Samples were evaluated using a FACSCalibur^TM^ flow cytometer (BD Biosciences, San Jose, CA, USA) equipped with a 488 nm argon ion laser and red diode (635 nm) laser. Fluorescent signals were acquired by Cell Quest Pro ™ software Version 6 (BD Biosciences, San Jose, CA, USA). The green FL1 channel using 530/30 nm band pass filter, orange FL2 channel using 585/42 nm band pass filter, and red FL3 channel using 670 nm long pass filter were used. Calibration was carried out periodically using standard calibration beads (BD CaliBRITE™; BD Biosciences, San Jose, CA, USA). At least, 10,000 events were measured in each sample. The gating strategy for the evaluation of obtained flow-cytometric data is shown in Figure 2.

### 2.5. Measurement of Haematological and Biochemical Parameters

Blood and serum sampling

Each animal in the experiment was healthy and in good condition. At the beginning of the experiment, control blood sampling took place on D0 (day zero), the next sampling was completed on day 30 (D30), the third sampling was completed on day 60 (D60), and the final blood sampling was completed on day 90 (D90). A qualified veterinarian extracted blood samples from the vena auricularis centralis and put them into two tubes. Samples were put into tubes without any additives for biochemical evaluation, while tubes with EDTA, as an anticoagulant, were intended for the haematological study. The blood serum, obtained from the blood using centrifugation at 1006× *g* for 20 min, was kept at −20 °C until further analysis.

Haematology analysis

The haematology parameters (WBC—total white blood cell count, LYM—lymphocytes count, MID—cell population of middle dimensions including monocytes and eosinophils, GRA—granulocytes count, LYM %—lymphocyte percentage, MID %—cell population of middle dimensions including monocytes and eosinophils percentage, GRA %—granulocytes percentage, RBC—red blood cell count, HGB—haemoglobin, HCT—haematocrit, MCV—mean corpuscular volume, MCH—mean corpuscular haemoglobin, MCHC—mean corpuscular haemoglobin concentration, RDWc—red cell distribution width calculated, RDWs—red cell distribution width standard, PLT—platelet count, PCT %—platelet percentage, MPV—mean platelet volume, PDWc—platelet distribution width calculated, and PDWs—standard) were analysed using an Abacus VET 5 haematology analyser. Each sample was measured three times, and the resulting average value was used for statistical evaluation.

Serum biochemistry analysis

The blood serum parameters (Ca—calcium, P—phosphorus, Mg—magnesium, TP—total proteins, albumin, AST—aspartate aminotransferase, ALT—alanine aminotransferase, ALP—alkaline phosphatase, Bili—direct bilirubin, Creat—enzymatic creatinine, Glu—glucose, Chol—cholesterol, TG—triglycerides, uric acid) were measured using a fully automated Randox RX Monaco biochemistry analyser (Randox Laboratories, Crumlin, UK) according to standard procedures recommended by the manufacturer. Serum ions (K^+^ and Cl^−^) were measured according to standard methodology using an EasyLyte electrolyte analyser (Medica, Bedford, MA, USA). The serum globulin (Glob) level was calculated by subtracting the serum albumin values from the total protein. The following formula was used to obtain the albumin-to-globulin ratio: A/G ratio = Albumin/(Total protein − Albumin)

### 2.6. Statistical Analysis

The experiments were performed on 30 sexually mature males (C = 10; E1= 10; E2 = 10). The obtained data were statistically analysed using STATGRAPHICS Centurion© (StatPoint Technologies, Inc., Warrenton, VA, USA). All data were checked for normality using a Kolmogorov–Smirnov test. Differences between monitored biomarkers according to individual samplings were evaluated using analysis of variance (ANOVA) with Tukey’s post hoc test and re-calculated using a Bonferroni test (all data are presented as mean ± standard deviation). The differences were considered significant at *p* < 0.05.

## 3. Results

### 3.1. Effect of Hempseed Cake on Weight Gains and Semen Quality

In our study, the hempseed cake at different concentrations was added to the normal rabbit diet. Although the animals were already physically and sexually adult, average weekly and total weight gains were recorded to find whether this additive has a negative effect on the nutrition and health of the animals. An insignificant decrease in body weight of animals fed with the hempseed cake supplement was observed (*p* > 0.05) (Table 3).

In our study, the sperm ejaculates collected from the experimental and control males were analysed for their quality. Thirty rabbit males were divided into three qualitatively homogeneous groups (C, E5, and E10; 10 males in each group) on the base of average sperm concentration, percentage of total motility, and progressively moving spermatozoa (Table 4, Table 5 and Table 6). The experiment was ongoing in the winter months from November to February. We found that differences in values of sperm concentration, sperm motility, and progressive motility between the groups were not statistically significant. In the last month, an improvement in all mentioned parameters was noted in all tested groups, which can be caused by seasonality and the onset of spring.

The viability, apoptotic-like changes, acrosomal status, mitochondrial activity, capacitation, and ROS of rabbit sperm were analysed by flow cytometry. We found that the differences between the individual groups were not statistically significant (*p* > 0.05; Table 7).

### 3.2. Haematology and Serum Biochemistry Observations

Before starting the administration of the experimental diet, the blood samples were taken for haematological and biochemical parameters. As shown in Table 8 and Table 9, the sets of animals chosen for individual experimental groups were consistent. Only the glucose parameter showed a difference between the C and E1 groups. However, we decided to keep the groups of animals in this composition for the next phase of the experiment due to the level of significance being close to 0.05 and the fact that these values were within the reference range according to the Merck Veterinary Manual. By comparison with the reference values, it is necessary to mention slightly increased values of RBC, MON (%), and Ca, as well as relatively high levels of phosphorus and ALP in all groups, whose associations with the experimental diet are followed in the next section.

The data obtained by the blood and serum analyses after one month of the experiment are presented in Table 10 and Table 11. Based on the statistical analysis, we found a statistically significant difference between the control group and the E1 group in the RDWs value (*p* < 0.05) for the haematological examination. During the biochemical examination, we noted a statistically significant increase in the phosphorus levels in the E5 and E10 groups compared to the control group (*p* < 0.01). No significant differences in other parameters were observed between the control and experimental groups. By comparing the determined values of the haematological parameters with the reference values, we noted a slightly increased MON % but also a decrease in the total content of platelets (PLT) in all groups of animals. The previous relatively high levels of P and ALP were in line with the reference values after the one-month diet.

Table 12 and Table 13 present the data gathered from the examination of the blood and serum during the experiment. The statistical analysis revealed a significant difference (*p* < 0.001) in the MCV parameter between the E10 and control groups and between E10 and E5. In comparison to the control group, we observed a decrease in MCH and RDWs (both *p* < 0.05) in the E10 group. PCT (%) changes were significantly different (*p* < 0.05) between E5 and E10 groups. The E10 group showed significantly lower chloride levels than the E5 group throughout the biochemical analysis (*p* < 0.01). A decreased tendency was also observed for the glucose content (C > E5 > E10; *p* < 0.001), along with a decrease in the levels of triglycerides and cholesterol when comparing E5 to control (*p* < 0.05). P and ALP levels were among the haematological and biochemical markers that were equivalent to reference values. The levels of Ca and MON (%) showed a slight increase, similar to the earlier sampling.

The effect of the experimental diet after three months on the monitored health indicators is presented in Table 14 and Table 15. We noted a decrease in a haematocrit in the E10 group compared to the control, as well as a decrease in platelets in both experimental groups (*p* < 0.01), which also caused the decrease in PCT % (*p* < 0.05). Biochemical analyses after three months of the experiment revealed an increase in phosphorus levels in E10 compared to the control (*p* < 0.05). Bilirubin and triglyceride levels had the opposite tendency; we noted a decrease in bilirubin in E2 compared to control (*p* < 0.05), as well as a decrease in TG in both E5 and E10 compared to control (*p* < 0.05). Phosphorus, calcium, and liver enzymes (especially ALP) levels remained normal after three months of experimental diet, which we consider to be positive especially for high concentrations of ALP before the administration of experimental diets.

## 4. Discussion

### 4.1. Effect of Hempseed Cake on Weight Gain

A promising direction in animal nutrition may be the use of alternative feeds containing bioactive compounds or mixtures of natural origin, phytoadditives or plant ex-tracts, probiotics, prebiotics, symbiotics, or oilseed by-products, such as hempseed cakes [22]. According to this author it may be more appropriate to apply the extract of the bioactive substance complex to livestock diets than addition of expellers or other forms of plant processing. Jacobson et al. [23] added hempseed meal to weaned Californian rabbits at four levels (CON, LOW, MED, and HIGH, representing 0, 25, 50, or 75% of the ration crude protein, respectively). Dry matter (DM) intake was greatest (*p* < 0.05) from LOW, followed by MED and then CON, and was least (*p* < 0.05) from HIGH. Similarly, total weight gain and average daily weight gain were greatest (*p* < 0.05) from LOW (1.230 g and 35.1 g/d, respectively), followed by MED (1.163 g and 33.2 g/d, respectively) and then CON (1.067 g and 30.5 g/d, respectively), and were least (*p* < 0.05) from HIGH (725 g and 20.7 g/d, respectively). DM digestibility was decreased by 10% from CON to LOW before increasing by 14% with HIGH. The variability of live weight gains in our study is due to different appetites in individual animals; in the groups where hemp was fed, the animals showed a lower appetite. At the beginning of the experiment, the rabbits had mostly balanced weight within the groups (3700–4200 g, average 3850 g). At the end of the experiment, they had (C: 4250–4900 g, average 4500 g), (E5: 3900–4800 g, average 4400 g), and (E10: 3800–4700 g, average 4350 g). This is the reason why the standard deviations were so large. In our experiment, there was a trend towards reduced weight gain in rabbits. This phenomenon was also observed by other authors [24,25], where hempseed oil in the quail diet significantly affected final body weight, feed conservation ratio, feed intake, and body weight gain. All performance parameters in the hempseed oil group were lower compared to the control group. In the study of Formelová et al. [26], no significant differences among the experimental groups in feed intake, body weight, and carcass parameters were found. Hempseed oil cake was included in rabbits’ diet with beneficial effect on carcass quality and enhanced the nutritional quality of rabbit meat with the focus on essential amino acids. Moreover, the data on volatile fatty acids (VFAs) show that the most intensive process was in the caecum of rabbits fed with this diet.

This effect may be due to the synergy of various compounds contained in *Cannabis* including more than 120 terpenoids, 100 cannabinoids, 50 hydrocarbons, 34 glycosidic compounds, 27 nitrogenous compounds, 25 non-cannabinoid phenols, 22 fatty acids, 21 simple acids, 18 amino acids, 13 simple ketones, 13 simple esters and lactones, 12 simple aldehydes, 11 proteins, glycoproteins, and enzymes, 11 steroids, nine trace elements, seven simple alcohols, two pigments, as well as vitamin K [27]. The endocannabinoid system is physiologically involved in regulation of appetite, pain, mood, memory, inflammation, insulin sensitivity, as well as fat and energy metabolism, with a wide variety of potential therapeutic implications for treatment of pain, neuropsychiatric disorders, neurological diseases, and inflammatory bowel, which may benefit from CB1 activation, as well as for treatment of obesity, type 2 diabetes, and hepatic or kidney disorders, which may benefit from CB1 antagonism [28].

Hempseed cake was also fed in other large and small farm animals like cattle [29,30,31], pigs [32], goats [33,34], and ewes [35,36]; however, it had no significant effect on weight gain or milk production. A reduction in average daily feed intake and average daily gain was also observed in broilers fed with 2.5% of hempseeds during the first 21 days of the treatment, while no difference was observed in body gain with diets at 4 and 7.5% of hempseeds [37]. Using cannabis at a level of 1.5% in the diet of broiler chickens could increase feed consumption relatively as well as increase weight at the end of the experiment [38].

### 4.2. Effect of Hempseed Cake on Semen Quality

The effects of different additives on rabbit sperm quality were reported [21,39,40,41,42]. Nevertheless, the quality of rabbit sperm after hempseed cake administration has not been described previously.

Except for sperm motility, it is important to evaluate other physiological changes in sperm. The flow-cytometric analyses include evaluation of different spermatozoa parameters, which influence the overall semen quality. Our results on sperm viability obtained using flow cytometry (Table 7) correspond to the data obtained by CASA, suggesting no deterioration in the sperm quality of in the experimental groups fed with hempseed cake. The common trait is sperm viability, which can be assessed through their plasma membrane integrity using SYBR-14 dye staining of live cells alone or in co-staining using dead cell dyes entering cells via damaged membranes, for example SYTOX Green dye [43,44]. YO-PRO-1 iodide is a nuclear dye which penetrates through the disrupted membrane and stains apoptotic cells [45]. Similarly, another apoptosis dye, Caspase 3/7, is used to detect apoptosis. Disruption of membrane integrity can cause damage to the acrosome. To detect this phenomenon, *Arachis hypogaea* (peanut) agglutinin (PNA) is usually used [46]. The increased mitochondrial activity is another important parameter of sperm activity. The quality of mitochondria can be assessed through membrane mitochondrial potential (MMP) by a fluorescent marker such as MitoTracker [47]. Capacitation level and the detection of intracellular Ca^2+^ in sperm can be measured using a FLUO-4 AM reagent [48]. Under physiological conditions, the level of ROS production is important for sperm functions, such as motility, viability, capacitation, and the acrosome reaction [49]. The cell-permeant CellROX dye appears interesting to detect the oxidative status in viable sperm [50]. In this study, we analysed sperm qualitative parameters through the fluorescent dyes. First, the following parameters were examined: live (SYBR-14+), apoptotic (YO-PRO-1+ and Caspase 3/7+), dead (SYTOX+), acrosome status (PNA+), mitochondrial activity (MitoTracker+), capacitation (FLUO-4+), and oxidative stress (CellROX+). Contrary to our primary hypothesis, we found no changes in all quality parameters resulting from hempseed cake administration. This is also in contrast with previous studies showing that administration of THC to animals declines sperm motility and viability as well as the acrosome reaction and also fertilization [51,52,53]. However, it is also known that activation of the CB1 receptor is required to analyse the effect of endocannabinoids [54]. As a result, it is possible that sperm quality parameters are only induced at concentrations substantially higher than those used in the present study.

### 4.3. Haematological and Biochemical Parameters

The primary goal before using these products in animal nutrition should be sufficient testing, primarily of overall health and reproductive potential [10,55,56,57,58]. Therefore, the haematology and serum biochemistry of rabbits fed hempseed cake should be monitored and compared with the reference values for healthy animals. Cozma et al. [33] evaluated the effect of hempseed oil on the milk characteristics of Carpathian goats for 31 days and did not observe significant differences in haematocrit, ALT, cholesterol, and triglycerides levels after one month of exposure, which corresponds to our results on increased serum phosphorus concentrations in the experimental groups, but within the reference values. In another study [59], the authors tested hempseed cake given at two concentrations (60 g/kg and 120 g/kg) to the diet of French Alpine goats for 45 days. They recorded similar tendencies, as in our study, when we compared the second experimental sampling date (after two months of experimental diet). It is primarily a decrease and subsequent increase in WBC and Mg levels, a significant decrease in MCV levels, an increase in MCHC, a decrease in total proteins, glucose, and triglycerides, and an increase in AST and ALT in the second experimental group. Different indices than in our study were recorded for cholesterol and calcium concentrations, however, with a minimal dose-dependent effect as well as with minimal deviations from reference values for all tested biochemical or haematological parameters. An interesting finding of their study is the decrease in enzymatic antioxidants (SOD and GPx) in the experimental groups, as the opposite effect would be expected due to the significant antioxidant potential and content of phenolic compounds in hempseed cake [10,60]. Hemp seed and hempseed oil are also outstanding sources of polyunsaturated fatty acids (PUFAs). The seeds contain approximately 30% of oil, with over 80% of it being composed of PUFAs [6,61]. The importance of omega-3 and omega-6 fatty acids as well as the correct ratio (n-6/n-3) has been described in many studies, confirming health protection, therapeutic effects, positive influence on the physiology of cells and tissues, protection against cardiovascular morbidity, anti-neuroinflammatory activity, or immunomodulatory effect [62,63,64,65,66]. Prociuk et al. [67] did not confirm the cardioprotective effects or the effect of hemp seed on body weight of rabbits during an eight-week experiment. However, they noted significantly reduced values of serum cholesterol and triglycerides in groups that consumed 10% ground hemp seed, 10% partially delipidated hemp seed, and 5% coconut oil, which corresponds to our findings. Other possible positive effects are longer life spans and improved cognitive function [68]. In their study, the authors used female mice fed an HB diet (two-thirds hemp seed + one-third *S. oleraceus*). In addition to the findings, the authors described in the HB group a protective effect against liver steatosis and hepatotoxicity, reduced spleen inflammatory and oxidative stress, improved gut microbiota, or increased insulin sensitivity, as well as comparable findings with our study (control vs. E5 group), decreased levels of plasma triglycerides, cholesterol, ALT, and AST. For other farm animals, partially positive effects on the health and enzymatic antioxidants of sows and piglets have been found [69,70]. Winders et al. [58], by determining the blood parameters of heifers, did not confirm the influence of the hempseed cake diet on the glucose content, which is in accordance with our findings after the first and third months of the hempseed cake diet. They also recorded increased plasma urea values and significant changes in plasma amino acid concentrations during 98 days of the diet. Several studies deal with the possible positive effect of this product on blood glucose levels in diabetic models [71]. Munteanu et al. [72] confirmed a significant decrease in hyperglycaemia in the diabetic hemp seeds feed group compared to the normal feeding group of diabetic Wistar rats. The most important confirmed agents of the hypoglycaemic activity are hemp protein peptides [73], which is demonstrably related to our findings.

Because hempseed cake is a high-protein co-product of hemp oil extraction, using it as a protein source in rabbit diets can lower the environmental impact and feed production costs. Additionally, hempseed cake contains a variety of phytochemicals that may be advantageous to rabbit performance and health, including flavonoids, terpenes, and cannabinoids. Nevertheless, this co-product also has some anti-nutritional components like lectins, phytates, and tannins [5,9,74], which may limit its inclusion level in diets. Because of this, the ideal concentration in rabbit diets must be chosen in a way that balances both its beneficial and detrimental effects. In our case, slightly positive effects on serum biochemistry (especially hepatic profile markers) in the E5 experimental group (5% hempseed cake) were observed. At a higher dose, in some cases the tendencies were opposite; however, by comparing haematological and biochemical markers with reference values during the entire experiment, we can state the safety of the diet even at this higher dose of hempseed cake. Nutritional quality, potential health benefits, or other biological effects offer a possibility to use this product for human health and nutrition.

## 5. Conclusions

This study explored the potential for using seed cake from hemp (*Cannabis sativa* L., variety Finola) as a protein feed for rabbits. Hempseed cake did not show a strict negative effect on the tested parameters in male rabbits. As a part of monitoring reproductive potential, we did not confirm the expected positive effect on oxidative stress parameters (previously described antioxidant potential of hempseed cake). Several deviations in health parameters were observed but within the reference values. These results are supported by biochemical and haematological findings. Since these findings were obtained using a small number of animals, they are of preliminary character. Therefore, this requires additional experiments using higher numbers of animals.

## Figures and Tables

**Figure 1 vetsci-11-00509-f001:**
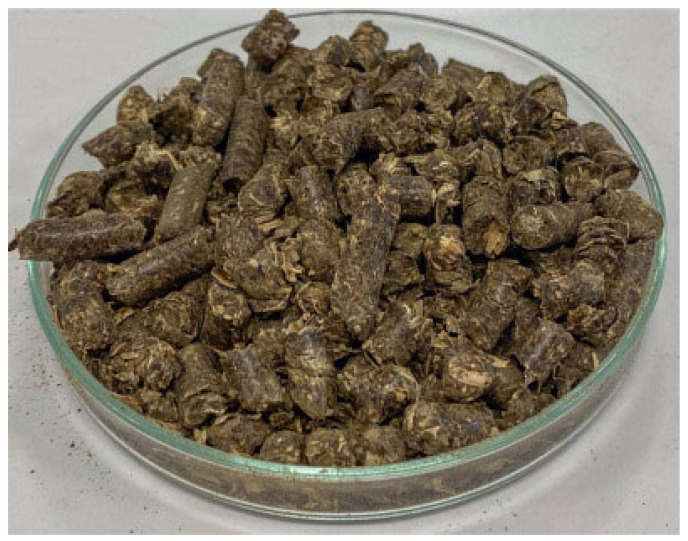
Hempseed cake originating from cold pressing hemp seeds to extract oil, produced at the Food Research Institute.

**Figure 2 vetsci-11-00509-f002:**
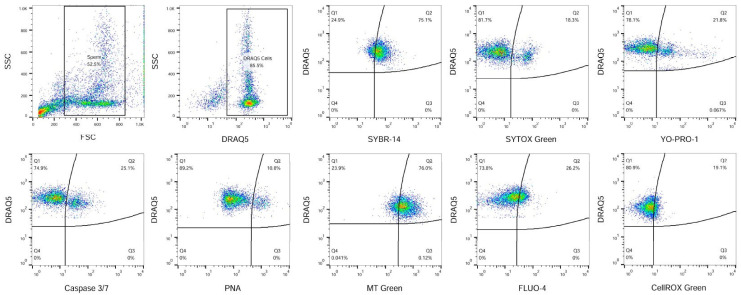
Illustrative flow-cytometric dot plots showing evaluation strategy of analysed fluorescent probes. Firstly, spermatozoa were gated using the FSC/SSC dot plot. DRAQ5 far-red dye was used to distinguish nucleated cells from other events. The specific sperm characteristics were evaluated from the DRAQ5 gated cells. Staining with SYBR-14 and SYTOX Green was used to evaluate viability. The proportion of apoptotic-like cells was evaluated using YO-PRO-1 and Caspase 3/7. Graph showing acrosome-damaged spermatozoa detected using PNA. Evaluation of rabbit sperm mitochondrial activity was using MT Green. Graph showing proportion of capacitated sperm which was detected using FLUO-4 and oxidatively damaged sperm using CellROX Green staining.

**Table 1 vetsci-11-00509-t001:** Composition of granulated diet for growing rabbits in original form.

Feed Ingredient Concentration (g/kg)	Control (Basal Diet)	E5 (with 5% Hempseed Cake)	EG10 (with 10% Hempseed Cake)
Lucerne meal	360	342	342
Extracted sunflower meal	55	52	52
Extracted rapeseed meal	55	52	52
Hempseed cake	0	50	100
Wheat bran	90	85	81
Oats	130	124	117
Malt sprouts	150	143	135
Maize	50	47.5	45
Sodium chloride	3	3	3
Mineral–vitamin mixture *	17	17	17
Barley grains	80	75	72
Limestone	10	10	9

E5—Basal diet with 5% hempseed cake; E10—Basal diet with 10% hempseed cake. * Commercial premix (KV; SIGI TRADE, Ltd. Slovak Republic).

**Table 2 vetsci-11-00509-t002:** Chemical composition of the hempseed cake and the granulated diets for growing rabbits.

Nutrients g·kg^−1^; * mg·kg^−1^	Hempseed Cake	Basal Diet	Basal Diet with 5% Hempseed Cake	Basal Diet with 10% Hempseed Cake
Crude protein	331.12	152.54	165.39	162.79
Crude fibre	286.17	156.63	162.39	153.99
Fat	113.87	28.13	33.97	33.05
Ash	58.04	77.31	80.14	84.16
Starch	10.85	188.29	189.00	181.62
Organic matter	841.78	810.21	817.41	821.71
ADF	325.33	145.53	182.19	185.43
NDF	382.52	334.36	334.08	328.44
Calcium	1.91	11.74	9.06	12.06
Phosphorus	7.57	5.89	6.34	6.82
Magnesium (Mg)	4.95	2.86	2.18	2.55
Sodium (Na)	0.23	1.31	1.61	1.23
Potassium (K)	8.60	10.17	10.37	9.82
Copper (Cu) *	23.03	22.32	23.14	21.18
Iron (Fe) *	150.16	503.63	624.49	577.00
ME MJ·kg^−1^	9.22	10.74	11.17	10.99

The vitamins/minerals premix provided per kg of complete diet: Vitamin A, 6000 IU; Vitamin D3, 1000 IU; Vitamin E, 50 mg; Vitamin B1, 1.7 mg; Vitamin B2, 8.0 mg; Vitamin B6, 3.0 mg; Vitamin B12, 0.01 mg; Vitamin K3, 0.5 mg; biotin, 0.2 mg; folic acid, 0.5 mg; nicotinic acid, 70 mg; choline chloride, 700 mg; Mn, 50 mg; Fe, 40 mg; Cu, 30 mg; Se, 0.2 mg. ADF—Acid detergent fibre; NDF—Neutral detergent fibre; ME—Metabolizable energy. *— nutrients are expressed in mg·kg^−1^.

**Table 3 vetsci-11-00509-t003:** Average weekly and total weight gains of the rabbit males.

	Weight Gain per Week (g)	Total Weight Gains (g)
C	271 ± 249	727 ± 555
E5	177 ± 131	533 ± 378
E10	166 ± 124	498 ± 305

Average weekly weight gains and total weight gains (g) of rabbit males fed hempseed cake supplemented into complete feed mixture. C—without dietary supplement of hempseed cake. E5—5 g hempseed cake/100 kg normal food. E10—10 g hempseed cake/100 kg normal food. Values represent means ± standard deviation (SD; *p* > 0.05). Differences at *p* < 0.05 were considered significant; number of samples (n = 10) per each group.

**Table 4 vetsci-11-00509-t004:** Sperm concentration.

Sperm Concentration (×10^9^)	D0	D30	D60	D90
C (n = 20)	1.47 ± 0.83	1.25 ± 0.32	0.82 ± 0.39	1.75 ± 0.69
E5 (n = 20)	1.55 ± 0.56	0.97 ± 0.32	0.89 ± 0.42	1.26 ± 0.70
E10 (n = 20)	1.12 ± 0.63	1.29 ± 0.98	0.97 ± 0.67	1.63 ± 1.28

Sperm concentration in rabbits fed with hempseed cake. C—no dietary hempseed cake. E5—5 g hempseed cake per 100 kg normal food. E10—10 g hempseed cake per 100 kg normal food. D0 (day zero), D30 (day 30), D60 (day 60), and D90 (day 90). Values represent means ± SD. Differences at *p* < 0.05 were considered significant; number of samples (n = 10) per each group.

**Table 5 vetsci-11-00509-t005:** Sperm total motility.

Sperm Motility (%)	D0	D30	D60	D90
C (n = 20)	50.0 ± 10.0	62.3 ± 11.7	49.7 ± 24.6	77.6 ± 5.25
E5 (n = 20)	49.8 ± 17.7	52.8 ± 12.8	54.4 ± 16.4	64.5 ± 17.8
E10 (n = 20)	50.5 ± 10.6	69.2 ± 13.5	65.6 ± 21.8	77.2 ± 11.3

Sperm motility in rabbits fed with hempseed cake. C—no dietary hempseed cake. E5—5 g hempseed cake per 100 kg normal food. E10—10 g hempseed cake per 100 kg normal food. D0 (day zero), D30 (day 30), D60 (day 60), and D90 (day 90). Values represent means ± SD. Differences at *p* < 0.05 were considered significant; number of samples (n = 10) per each group.

**Table 6 vetsci-11-00509-t006:** Sperm progressive motility.

Sperm Progressive Motility (%)	D0	D30	D60	D90
C (n = 20)	31.1 ± 12.3	42.7 ± 17.4	33.5 ± 20.6	60.3 ± 9.1
E5 (n = 20)	31.2 ± 20.1	28.3 ± 19.5	37.2 ± 19.8	51.3 ± 16.8
E10 (n = 20)	33.2 ± 19.8	46.6 ± 22.4	44.6 ± 26.1	55.6 ± 24.7

Sperm progressive motility in rabbits fed with hempseed cake. C—no dietary hempseed cake. E5—5 g hempseed cake per 100 kg normal food. E10—10 g hempseed cake per 100 kg normal food. D0 (day zero), D30 (day 30), D60 (day 60), and D90 (day 90). Values represent means ± SD. Differences at *p* < 0.05 were considered significant; number of samples (n = 10) per each group.

**Table 7 vetsci-11-00509-t007:** Spermatozoa characteristics using flow cytometry, %.

Viability			Apoptosis-like Changes		Acrosomal Status	MA	Capacitation	ROS
Sample	SYBR-14^+^	SYTOX Green^+^	YoPro-1^+^	Caspase 3/7^+^	PNA^+^	MT Green^+^	FLUO-4^+^	CellROX^+^
C	83.8 ± 7.7	15.0 ± 3.4	17.7 ± 3.6	21.7 ± 2.93	11.8 ± 3.6	57.6 ± 16.5	16.6 ± 8.6	16.8 ± 13.8
E5	79.9 ± 1.3	21.7 ± 11.	26.4 ± 6.2	25.0 ± 10.2	19.5 ± 7.6	58.4 ± 11.7	15.1 ± 8.5	19.3 ± 13.6
E10	83.4 ± 1.7	16.6 ± 0.8	16.5 ± 3.5	10.8 ± 1.65	10.3 ± 5.4	58.3 ± 17.3	15.9 ± 6.1	23.9 ± 15.3

MA—Mitochondrial activity, ROS—reactive oxygen species. Values represent means ± SD. Differences at *p* < 0.05 were considered significant; number of samples (n = 10) per each group.

**Table 8 vetsci-11-00509-t008:** Haematological examination of experimental animals assigned to groups before administration of the experimental diet (results are presented as mean ± SD).

Haematological Parameter	C (10)	E5 (10)	E10 (10)	*p*-Value	Reference Ranges ^1^
WBC (10^9^·L^−1^)	9.56 ± 1.72	8.37 ± 2.98	9.59 ± 1.23	ns	6–12
LYM (10^9^·L^−1^)	2.38 ± 1.96	3.82 ± 2.37	4.00 ± 1.42	ns	1.6–10.6
MON (10^9^·L^−1^)	0.48 ± 0.31	0.38 ± 0.38	0.45 ± 0.14	ns	0.05–0.5
NEU (10^9^·L^−1^)	6.70 ± 3.08	4.18 ± 3.13	5.15 ± 1.93	ns	1–9.4
LYM (%)	27.6 ± 23.13	47.6 ± 23.7	42.4 ± 16.9	ns	30–85
MON (%)	4.85 ± 2.44	4.10 ± 2.40	4.70 ± 1.70	ns	1–4
NEU (%)	67.5 ± 21.8	48.2 ± 22.4	52.8 ± 16.0	ns	20–60
RBC (10^12^·L^−1^)	6.59 ± 0.53	7.58 ± 1.64	7.24 ± 0.63	ns	4–7
HGB (g·dL^−1^)	13.3 ± 0.81	12.9 ± 2.38	14.1 ± 0.67	ns	8–15
HCT (%)	39.64 ± 2.19	44.29 ± 8.79	41.83 ± 1.96	ns	-
MCV (fL)	60.4 ± 2.92	58.7 ± 2.31	58.0 ± 3.60	ns	58–67
MCH (pg)	20.2 ± 0.80	17.9 ± 4.56	19.5 ± 1.27	ns	17.1–23.5
MCHC (g·dL^−1^)	33.5 ± 0.48	29.6 ± 7.95	33.6 ± 0.18	ns	29–37
RDWc (%)	15.4 ± 0.49	16.1 ± 1.27	15.6 ± 0.55	ns	-
RDWs (fL)	37.8 ± 1.32	38.1 ± 2.10	36.5 ± 1.23	ns	-
PLT (10^9^·L^−1^)	245 ± 96.3	264 ± 64.9	278 ± 47.6	ns	250–650
MPV (fl)	6.03 ± 0.40	5.98 ± 0.28	6.18 ± 0.25	ns	-
PCT (%)	0.14 ± 0.05	0.16 ± 0.04	0.17 ± 0.03	ns	-
PDWc (%)	29.1 ± 1.67	29.1 ± 1.60	29.9 ± 1.14	ns	-
PDWS (fl)	7.12 ± 0.93	7.14 ± 0.94	7.50 ± 0.63	ns	-

ns—not significant; ^1^ MSD Merck Veterinary Manual (Modified Nov 2022)—Haematology Reference Ranges. (WBC—total white blood cell count, LYM—lymphocytes count, LYM %—lymphocyte percentage, RBC—red blood cell count, HGB—haemoglobin, HCT—haematocrit, MCV—mean corpuscular volume, MCH—mean corpuscular haemoglobin, MCHC—mean corpuscular haemoglobin concentration, RDWc—red cell distribution width calculated, RDWs—red cell distribution width standard, PLT—platelet count, PCT %—platelet percentage, MPV—mean platelet volume, PDWc—platelet distribution width, and PDWs—platelet distribution width standard). number of samples (n = 10) per each group.

**Table 9 vetsci-11-00509-t009:** Serum biochemistry examination of experimental animals assigned to groups before administration of the experimental diet (results are presented as mean ± SD).

Biochemistry Parameter	C (10)	E5 (10)	E10 (10)	*p*-Value	Reference Ranges ^1^
K (mM·L^−1^)	5.16 ± 0.68	5.33 ± 0.65	5.30 ± 0.45	ns	3.5–6.9
Cl (mM·L^−1^)	102 ± 1.45	102 ± 2.42	99.6 ± 4.36	ns	-
Ca (mM·L^−1^)	3.53 ± 0.18	3.65 ± 0.18	3.79 ± 0.22	ns	2.7–3.5
P (mM·L^−1^)	4.90 ± 0.91	4.84 ± 1.14	5.33 ± 0.45	ns	1.3–2.1
Mg (mM·L^−1^)	1.03 ± 0.09	1.10 ± 0.11	1.12 ± 0.13	ns	-
Albumin (g·L^−1^)	39.8 ± 8.26	45.2 ± 4.01	46.6 ± 4.90	ns	36–48
ALP (U·L^−1^)	114 ± 44.6	139 ± 20.7	120 ± 41.3	ns	41–92
ALT (U·L^−1^)	45.4 ± 14.2	60.3 ± 45.3	67.4 ± 22.6	ns	45–80
AST (U·L^−1^)	39.61 ± 18.1	37.88 ± 8.68	45.56 ± 6.41	ns	35–130
Bili (µM·L^−1^)	1.27 ± 0.76	1.58 ± 0.57	1.46 ± 0.69	ns	0–12
Creat (µM·L^−1^)	62.6 ± 8.32	71.9 ± 23.2	60.2 ± 12.4	ns	44.2–221
Glu (mM·L^−1^)	7.86 ± 0.48 ^a^	8.56 ± 0.59 ^b^	8.38 ± 0.50 ^a,b^	0.0368	4.1–8.6
Chol. (mM·L^−1^)	0.90 ± 0.40	1.10 ± 0.30	1.15 ± 0.49	ns	0.3–2.1
TP (g·L^−1^)	70.3 ± 15.2	74.3 ± 14.9	74.8 ± 12.7	ns	54–75
TG (mM·L^−1^)	0.65 ± 0.32	0.79 ± 0.33	0.82 ± 0.34	ns	-
Uric acid (µM·L^−1^)	3.56 ± 2.47	2.61 ± 2.20	4.19 ± 1.14	ns	-
Globulin (g·L^−1^)	30.5 ± 16.2	29.0 ± 12.4	28.2 ± 9.34	ns	16–29
Alb/Glob ratio	1.77 ± 1.15	1.81 ± 0.72	1.85 ± 0.75	ns	-

ns—not significant; ^1^ MSD Merck Veterinary Manual (Modified Nov 2022)—Serum Biochemical Analysis Reference Ranges. (Ca—calcium, P—phosphorus, Mg—magnesium, TP—total proteins, albumin, AST—aspartate aminotransferase, ALT—alanine aminotransferase, ALP—alkaline phosphatase, Bili—direct bilirubin, Creat—enzymatic creatinine, Glu—glucose, Chol—cholesterol, TG—triglycerides, uric acid). ^a,b^ Values within a row with different superscripts differ significantly; number of samples (n = 10) per each group.

**Table 10 vetsci-11-00509-t010:** Haematological evaluation of animals after one month of experimental diet (results are presented as mean ± SD).

Haematological Parameter	C (10)	E5 (10)	E10 (10)	*p*-Value	Reference Ranges ^1^
WBC (10^9^·L^−1^)	7.64 ± 1.14	8.25 ± 2.38	10.374.71	ns	6–12
LYM (10^9^·L^−1^)	2.56 ± 0.96	2.96 ± 2.05	3.11 ± 1.38	ns	1.6–10.6
MON (10^9^·L^−1^)	0.42 ± 0.13	0.51 ± 0.18	0.29 ± 0.22	ns	0.05–0.5
NEU (10^9^·L^−1^)	4.65 ± 1.15	4.78 ± 3.33	6.84 ± 4.64	ns	1–9.4
LYM (%)	33.78±13.45	39.2 ± 31.16	34.83±21.36	ns	30–85
MON (%)	5.48 ± 1.21	6.48 ± 2.60	4.23 ± 2.04	ns	1–4
NEU (%)	60.7 ± 12.9	54.4 ± 31.8	60.9 ± 20.2	ns	20–60
RBC (10^12^·L^−1^)	6.18 ± 0.19	6.29 ± 0.59	5.92 ± 0.81	ns	4–7
HGB (g·dL^−1^)	13.3 ± 0.37	12. ± 1.14	12.1 ± 1.18	ns	8–15
HCT (%)	37.2 ± 1.02	36.4 ± 3.18	34.8 ± 2.97	ns	-
MCV (fL)	60.2 ± 2.22	57.9 ± 2.35	59.3 ± 3.20	ns	58–67
MCH (pg)	21.5 ± 0.82	20.2 ± 0.92	20.5 ± 0.97	ns	17.1–23.5
MCHC (g·dL^−1^)	35.6 ± 0.49	34.9 ± 0.59	34.7 ± 1.04	ns	29–37
RDWc (%)	15.6 ± 0.53	15.4 ± 0.69	16.1 ± 0.97	ns	-
RDWs (fL)	38.3 ± 1.47 ^a^	36.1 ± 0.69 ^b^	38.3 ± 2.30 ^a,b^	0.0200	-
PLT (10^9^·L^−1^)	218. ± 58.3	221 ± 48.5	218 ± 74.7	ns	250–650
MPV (fl)	5.65 ± 0.21	5.90 ± 0.36	5.98 ± 0.24	ns	-
PCT (%)	0.12 ± 0.03	0.12 ± 0.04	0.13 ± 0.04	ns	-
PDWc (%)	27.8 ± 1.71	28.8 ± 1.15	29.4 ± 1.52	ns	-
PDWS (fl)	6.50 ± 0.79	6.97 ± 0.57	7.28 ± 0.79	ns	-

ns—not significant; values with different superscript letters in a same row are significantly different; ^1^ MSD Merck Veterinary Manual (Modified Nov 2022)—Haematology Reference Ranges. (WBC—total white blood cell count, LYM—lymphocytes count, LYM %—lymphocyte percentage, RBC—red blood cell count, HGB—haemoglobin, HCT—haematocrit, MCV—mean corpuscular volume, MCH—mean corpuscular haemoglobin, MCHC—mean corpuscular haemoglobin concentration, RDWc—red cell distribution width calculated, RDWs—red cell distribution width standard, PLT—platelet count, PCT %—platelet percentage, MPV—mean platelet volume, PDWc—platelet distribution width calculated, and PDWs—platelet distribution width standard). ^a,b^ Values within a row with different superscripts differ significantly; number of samples (n = 10) per each group.

**Table 11 vetsci-11-00509-t011:** Serum biochemistry evaluation of animals after one month of experimental diet (results are presented as mean ± SD).

Biochemical Parameter	C (10)	E5 (10)	E10 (10)	*p*-Value	Reference Ranges ^1^
K (mM·L^−1^)	4.37 ± 0.37	4.54 ± 0.51	4.48 ± 0.36	ns	3.5–6.9
Cl (mM·L^−1^)	104. ± 1.77	104 ± 2.26	105 ± 1.02	ns	-
Ca (mM·L^−1^)	3.68 ± 0.11	3.58 ± 0.29	3.47 ± 0.12	ns	2.7–3.5
P (mM·L^−1^)	1.34 ± 0.14 ^a^	1.55 ± 0.15 ^b^	1.55 ± 0.13 ^b^	0.0072	1.3–2.1
Mg (mM·L^−1^)	1.06 ± 0.05	1.04 ± 0.18	1.06 ± 0.05	ns	-
Albumin (g·L^−1^)	47.7 ± 1.76	46.9 ± 5.48	46.1 ± 2.78	ns	36–48
ALP (U·L^−1^)	77.0 ± 28.5	73.9 ± 25.4	69.7 ± 34.2	ns	41–92
ALT (U·L^−1^)	51.6 ± 13.5	42.7 ± 12.2	54.2 ± 13.7	ns	45–80
AST (U·L^−1^)	28.8 ± 6.91	27.645.75	29.3 ± 7.08	ns	35–130
Bili (µM·L^−1^)	0.85 ± 0.23	0.73 ± 0.22	0.93 ± 0.25	ns	0–12
Creat (µM·L^−1^)	54.6 ± 10.5	57.1 ± 14.5	52.1 ± 9.03	ns	44.2–221
Glu (mM·L^−1^)	7.88 ± 0.32	8.03 ± 0.79	8.02 ± 0.76	ns	4.1–8.6
Chol. (mM·L^−1^)	0.68 ± 0.28	0.77 ± 0.76	0.69 ± 0.19	ns	0.3–2.1
TP (g·L^−1^)	69.0 ± 4.08	70.3 ± 4.26	66.3 ± 2.01	ns	54–75
TG (mM·L^−1^)	1.23 ± 1.09	1.03 ± 0.74	1.33 ± 0.51	ns	-
Uric acid (µM·L^−1^)	3.28 ± 3.03	2.19 ± 2.18	1.71 ± 1.17	ns	-
Globulin (g·L^−1^)	21.3 ± 4.97	23.5 ± 8.62	20.2 ± 2.57	ns	16–29
Alb/Glob ratio	2.34 ± 0.52	2.24 ± 0.79	2.32 ± 0.39	ns	-

ns—not significant; values with different superscript letters in a same row are significantly different; ^1^ MSD Merck Veterinary Manual (Modified Nov 2022)—Haematology Reference Ranges (Ca—calcium, P—phosphorus, Mg—magnesium, TP—total proteins, albumin, AST—aspartate aminotransferase, ALT—alanine aminotransferase, ALP—alkaline phosphatase, Bili—direct bilirubin, Creat—enzymatic creatinine, Glu—glucose, Chol—cholesterol, TG—triglycerides, uric acid). ^a,b^ Values within a row with different superscripts differ significantly; number of samples (n = 10) per each group.

**Table 12 vetsci-11-00509-t012:** Haematological evaluation of animals after two months of experimental diet (results are presented as mean ± SD).

Haematological Parameter	C (10)	E5 (10)	E10 (10)	*p*-Value	Reference Ranges ^1^
WBC (10^9^·L^−1^)	10.0 ± 2.83	7.67 ± 2.79	10.4 ± 2.12	ns	6–12
LYM (10^9^·L^−1^)	4.62 ± 1.08	4.77 ± 2.35	4.15 ± 1.66	ns	1.6–10.6
MON (10^9^·L^−1^)	0.43 ± 0.33	0.42 ± 0.19	0.51 ± 0.16	ns	0.05–0.5
NEU (10^9^·L^−1^)	4.95 ± 3.37	2.48 ± 2.44	5.62 ± 2.87	ns	1–9.4
LYM (%)	49.8 ± 18.9	62.6 ± 19.5	42.7 ± 19.8	ns	30–85
MON (%)	4.52 ± 4.05	5.59 ± 1.59	4.90 ± 0.97	ns	1–4
NEU (%)	45.6 ± 40.0	31.8 ± 19.4	52.4 ± 18.9	ns	20–60
RBC (10^12^·L^−1^)	6.26 ± 6.21	6.53 ± 0.32	6.55 ± 0.67	ns	4–7
HGB (g·dL^−1^)	13.1 ± 13.05	13.2 ± 0.75	12.70 ± 1.51	ns	8–15
HCT (%)	39.9 ± 39.0	39.6 ± 2.47	37.6 ± 4.08	ns	-
MCV (fL)	63.8 ± 63.0 ^a^	60.8 ± 2.49 ^a^	57.4 ± 2.77 ^b^	0.0003	58–67
MCH (pg)	20.9 ± 20.90 ^a^	20.2 ± 0.76 ^a,b^	19.4 ± 1.34 ^b^	0.0300	17.1–23.5
MCHC (g·dL^−1^)	32.82 ± 32.65	33.23 ± 0.87	33.78 ± 1.08	ns	29–37
RDWc (%)	15.4 ± 15.4	15.4 ± 1.24	15.5 ± 0.94	ns	-
RDWs (fL)	40.3 ± 3.94 ^a^	36.8 ± 1.56 ^a,b^	36.0 ± 2.03 ^b^	0.0175	-
PLT (10^9^·L^−1^)	301 ± 58.3	232 ± 67.6	323 ± 101	ns	250–650
MPV (fl)	5.82 ± 0.25	5.78 ± 0.25	5.97 ± 0.46	ns	-
PCT (%)	0.18 ± 0.04 ^a,b^	0.14 ± 0.04 ^a^	0.19 ± 0.05 ^b^	0.0495	-
PDWc (%)	28.4 ± 1.54	28.6 ± 1.39	29.4 ± 1.33	ns	-
PDWS (fl)	6.75 ± 0.72	6.86 ± 0.67	7.23 ± 0.69	ns	-

ns—not significant; values with different superscript letters in a same row are significantly different; ^1^ MSD Merck Veterinary Manual (Modified Nov 2022)—Haematology Reference Ranges. (WBC—total white blood cell count, LYM—lymphocytes count, LYM %—lymphocyte percentage, RBC—red blood cell count, HGB—haemoglobin, HCT—haematocrit, MCV—mean corpuscular volume, MCH—mean corpuscular haemoglobin, MCHC—mean corpuscular haemoglobin concentration, RDWc—red cell distribution width calculated, RDWs—red cell distribution width standard, PLT—platelet count, PCT %—platelet percentage, MPV—mean platelet volume, PDWc—platelet distribution width, and PDWs—platelet distribution width standard). ^a,b^ Values within a row with different superscripts differ significantly; number of samples (n = 10) per each group.

**Table 13 vetsci-11-00509-t013:** Serum biochemistry evaluation of animals after two months of experimental diet (results are presented as mean ± SD).

Biochemical Parameter	C (10)	E5 (10)	E10 (10)	*p*-Value	Reference Ranges ^1^
K (mM·L^−1^)	5.00 ± 0.65	4.45 ± 0.44	4.63 ± 0.36	ns	3.5–6.9
Cl (mM·L^−1^)	104 ± 1.29 ^a,b^	107 ± 2.07 ^a^	102 ± 3.6 ^b^	0.0030	-
Ca (mM·L^−1^)	4.57 ± 1.98	3.67 ± 0.16	3.60 ± 0.13	ns	2.7–3.5
P (mM·L^−1^)	1.29 ± 0.11	1.46 ± 0.33	1.29 ± 0.22	ns	1.3–2.1
Mg (mM·L^−1^)	1.00 ± 0.16	0.95 ± 0.04	1.02 ± 0.08	ns	-
Albumin (g·L^−1^)	36.2 ± 2.79	37.6 ± 1.51	34.9 ± 3.02	ns	36–48
ALP (U·L^−1^)	53.6 ± 11.9	65.8 ± 12.0	59.9 ± 25.0	ns	41–92
ALT (U·L^−1^)	29.9 ± 7.01	23.9 ± 5.63	30.4 ± 8.07	ns	45–80
AST (U·L^−1^)	30.0 ± 6.43	26.1 ± 3.57	33.9 ± 10.5	ns	35–130
Bili (µM·L^−1^)	1.24 ± 0.43	0.90 ± 0.21	1.10 ± 0.21	ns	0–12
Creat (µM·L^−1^)	62.6 ± 13.9	62.9 ± 9.09	61.7 ± 10.9	ns	44.2–221
Glu (mM·L^−1^)	7.72 ± 0.47 ^a^	6.73 ± 0.57 ^b^	5.78 ± 0.59 ^c^	0.0000	4.1–8.6
Chol. (mM·L^−1^)	0.93 ± 0.36 ^a^	0.53 ± 0.17 ^b^	0.79 ± 0.34	0.0379	0.3–2.1
TP (g·L^−1^)	69.2 ± 2.76	69.5 ± 6.94	64.9 ± 5.27	ns	54–75
TG (mM·L^−1^)	1.72 ± 1.24 ^a^	0.72 ± 0.35 ^b^	1.03 ± 0.26 ^a,b^	0.0311	-
Uric acid (µM·L^−1^)	7.84 ± 2.92	7.66 ± 2.52	7.92 ± 2.92	ns	-
Globulin (g·L^−1^)	32.9 ± 4.08	31.8 ± 7.98	30.0 ± 4.65	ns	16–29
A/G ratio	1.12 ± 0.18	1.23 ± 0.23	1.18 ± 0.20	ns	-

ns—not significant; values with different superscript letters in a same row are significantly different; ^1^ MSD Merck Veterinary Manual (Modified Nov 2022)—Haematology Reference Ranges. (Ca—calcium, P—phosphorus, Mg—magnesium, TP—total proteins, albumin, AST—aspartate aminotransferase, ALT—alanine aminotransferase, ALP—alkaline phosphatase, Bili—direct bilirubin, Creat—enzymatic creatinine, Glu—glucose, Chol—cholesterol, TG—triglycerides, uric acid). ^a,b^ Values within a row with different superscripts differ significantly; number of samples (n = 10) per each group.

**Table 14 vetsci-11-00509-t014:** Haematological evaluation of animals after three months of experimental diet (results are presented as mean ± SD).

Haematological Parameter	C (10)	E5 (10)	E10 (10)	*p*-Value	Reference Ranges ^1^
WBC (10^9^·L^−1^)	10.5 ± 3.48	6.84 ± 0.91	7.32 ± 3.53	ns	6–12
LYM (10^9^·L^−1^)	4.97 ± 1.46	4.84 ± 1.20	3.35 ± 1.78	ns	1.6–10.6
MON (10^9^·L^−1^)	0.54 ± 0.15	0.37 ± 0.17	0.47 ± 0.30	ns	0.05–0.5
NEU (10^9^·L^−1^)	4.91 ± 2.53	1.62 ± 1.61	3.51 ± 4.05	ns	1–9.4
LYM (%)	48.4 ± 10.21	72.6 ± 22.2	52.7 ± 22.4	ns	30–85
MON (%)	5.57 ± 1.75	5.30 ± 1.90	6.10 ± 1.74	ns	1–4
NEU (%)	46.1 ± 10.75	22.1 ± 21.1	41.2 ± 21.8	ns	20–60
RBC (10^12^·L^−1^)	7.16 ± 1.07	6.59 ± 0.65	5.76 ± 1.32	ns	4–7
HGB (g·dL^−1^)	12.2 ± 1.81	12.7 ± 1.16	10.5 ± 2.40	ns	8–15
HCT (%)	43.2 ± 6.56 ^a^	39.9 ± 3.21 ^a,b^	33.5 ± 7.21 ^b^	0.0175	-
MCV (fL)	60.2 ± 4.12	60.7 ± 1.38	58.7 ± 3.71	ns	58–67
MCH (pg)	17.5 ± 3.80	19.3 ± 0.52	18.3 ± 0.96	ns	17.1–23.5
MCHC (g·dL^−1^)	30.6 ± 1.82	31.8 ± 0.71	31.3 ± 0.95	ns	29–37
RDWc (%)	15.3 ± 1.07	15.3 ± 0.53	15.9 ± 0.79	ns	-
RDWs (fL)	38.2 ± 2.97	37.9 ± 1.57	37.9 ± 2.07	ns	-
PLT (10^9^·L^−1^)	336 ± 112 ^a^	185 ± 72.9 ^b^	193 ± 41.9 ^b^	0.0063	250–650
MPV (fl)	5.82 ± 0.28	5.92 ± 0.15	6.13 ± 0.40	ns	-
PCT (%)	0.22 ± 0.12 ^a^	0.11 ± 0.04 ^b^	0.10 ± 0.05 ^b^	0.0141	-
PDWc (%)	28.3 ± 0.96	28.9 ± 1.58	30.1 ± 1.76	ns	-
PDWS (fl)	6.68 ± 0.47	7.02 ± 0.84	7.64 ± 0.96	ns	-

ns—not significant; values with different superscript letters in a same row are significantly different; ^1^ MSD Merck Veterinary Manual (Modified Nov 2022)—Haematology Reference Ranges. (WBC—total white blood cell count, LYM—lymphocytes count, LYM %—lymphocyte percentage, RBC—red blood cell count, HGB—haemoglobin, HCT—haematocrit, MCV—mean corpuscular volume, MCH—mean corpuscular haemoglobin, MCHC—mean corpuscular haemoglobin concentration, RDWc—red cell distribution width calculated, RDWs—red cell distribution width standard, PLT—platelet count, PCT %—platelet percentage, MPV—mean platelet volume, PDWc—platelet distribution width calculated, and PDWs—platelet distribution width standard). ^a,b^ Values within a row with different superscripts differ significantly; number of samples (n = 10) per each group.

**Table 15 vetsci-11-00509-t015:** Serum biochemistry evaluation of animals after three months of experimental diet (results are presented as mean ± SD).

Biochemical Parameter	C (10)	E5 (10)	E10 (10)	*p*-Value	Reference Ranges ^1^
K (mM·L^−1^)	5.02 ± 0.43	4.75 ± 0.48	4.65 ± 0.22	ns	3.5–6.9
Cl (mM·L^−1^)	102 ± 3.81	103 ± 4.63	105 ± 1.58	ns	-
Ca (mM·L^−1^)	3.56 ± 0.19	3.46 ± 0.17	3.49 ± 0.13	ns	2.7–3.5
P (mM·L^−1^)	1.08 ± 0.16 ^a^	1.21 ± 0.12 ^a,b^	1.27 ± 0.15 ^b^	0.0431	1.3–2.1
Mg (mM·L^−1^)	0.98 ± 0.13	0.92 ± 0.07	0.98 ± 0.07	ns	-
Albumin (g·L^−1^)	33.8 ± 2.77	35.3 ± 3.08	35.5 ± 2.80	ns	36–48
ALP (U·L^−1^)	41.9 ± 12.6	49.1 ± 19.4	55.3 ± 18.3	ns	41–92
ALT (U·L^−1^)	28.7 ± 7.58 ^a,b^	23.2 ± 5.35 ^a^	35.3 ± 13.7 ^b^	0.0460	45–80
AST (U·L^−1^)	30.4 ± 7.01	30.2 ± 9.46	32.5 ± 13.2	ns	35–130
Bili (µM·L^−1^)	1.13 ± 0.14 ^a^	0.87 ± 0.18 ^b^	0.97 ± 0.20 ^a,b^	0.0210	0–12
Creat (µM·L^−1^)	60.1 ± 9.93	59.0 ± 9.31	59.5 ± 14.0	ns	44.2–221
Glu (mM·L^−1^)	7.32 ± 0.36	7.31 ± 0.52	7.07 ± 0.35	ns	4.1–8.6
Chol. (mM·L^−1^)	0.81 ± 0.57	0.51 ± 0.16	0.75 ± 0.43	ns	0.3–2.1
TP (g·L^−1^)	66.2 ± 5.26	66.0 ± 7.48	65.2 ± 2.60	ns	54–75
TG (mM·L^−1^)	1.61 ± 0.86 ^a^	0.85 ± 0.34 ^b^	0.91 ± 0.28 ^b^	0.0160	-
Uric acid (µM·L^−1^)	6.83 ± 2.88	7.32 ± 1.62	7.18 ± 1.78	ns	-
Globulin (g·L^−1^)	32.4 ± 6.15	30.8 ± 7.04	29.7 ± 4.25	ns	16–29
Alb/Glob ratio	1.08 ± 0.21	1.19 ± 0.24	1.22 ± 0.23	ns	-

ns—not significant; values with different superscript letters in a same row are significantly different; ^1^ MSD Merck Veterinary Manual (Modified Nov 2022)—Haematology Reference Ranges. (Ca—calcium, P—phosphorus, Mg—magnesium, TP—total proteins, albumin, AST—aspartate aminotransferase, ALT—alanine aminotransferase, ALP—alkaline phosphatase, Bili—direct bilirubin, Creat—enzymatic creatinine, Glu—glucose, Chol—cholesterol, TG—triglycerides, uric acid). ^a,b^ Values within a row with different superscripts differ significantly; number of samples (n = 10) per each group.

## Data Availability

The data presented in this study are available in the article.

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
