# Peer review of "The Effects of Adding Hempseed Cake on Sperm Traits, Body Weight, Haematological and Biochemical Parameters in Rabbit Males"

_vetsci, 2024, doi:10.3390/vetsci11100509_

Round 1
Reviewer 1 Report (Previous Reviewer 2)
Comments and Suggestions for Authors
1. Introduction
Still pending:
The importance of rabbit production worldwide should be further developed, using for example the FAOSTAT references.
2.2 animals
Still pending:
The quantity of the water is not indicated. Ad libitum?
Author Response
Response to Reviewer 1 Comments
Dear rewiever
Thank you for your re-reading and rest remarks.
- Introduction
Still pending:
The importance of rabbit production worldwide should be further developed, using for example the FAOSTAT references.
Response 1: We accepted the reviewers' comment and we added FAOSTAT the reference.
Lebas F, Coudert P, de Rochambeau H, Thébault Rg. The rabbit husbandry, health and reproduction. FAO Animal Pruduction and Health Series 1997;21:205p. ISSN 1010-9021.
Point 2: The quantity of the water is not indicated. Ad libitum?
Response 2: The water ws served ad libitum. It was added to the text.
Thank you once again for your time to improve the quality of our manuscript.

Reviewer 2 Report (Previous Reviewer 3)
Comments and Suggestions for Authors
The manuscript was significantely improved. Thank you for considering the reviewers comments.
By re-reading, I still have a few recommandations howerver.
L246: add "numerical" between "slight" and "decrease"
Table 3: add the P values
Tables 4, 5 and 6: add the units and the P values.
Table 7: add P values
Disucussion first paragraph: In table 3 the reader see a huge variation that may explain the absence of significance. The discussion needs to address the reasons for this important variation in the weight gain results. Acutally the discussion does not relate the other publication to the present study.
Thank you
Author Response
Response to Reviewer 2 Comments
Dear rewiever
Thank you for your re-reading and good and inspirating remarks.
Point 1: L246: add "numerical" between "slight" and "decrease"
Response 1: We accepted the reviewers' comment and the sentence was rewritten.
Point 2: Table 3: add the P values.
Response 2: The P values were added.
Point 3: Tables 4, 5 and 6: add the units and the P values.
Response 3: The P values and units were added.
Point 4: Table 7: add the P values.
Response 4: The P values were added.
Point 5: Disucussion first paragraph: In table 3 the reader see a huge variation that may explain the absence of significance. The discussion needs to address the reasons for this important variation in the weight gain results. Acutally the discussion does not relate the other publication to the present study.
Response 5: The variability of live weight gains in our study is due to different appetites in individual animals; in the groups where hemp was fed, the animals showed a lower appetite. At the beginning of the experiment, the rabbits had mostly balanced weight within the groups (3700-4200g, average 3850g). At the end of the experiment, they had (C: 4250-4900g, average 4500g), (E5: 3900-4800g, average 4400g) and (E10: 3800-4700g, average 4350g). This is the reason why the standard deviations were so large. In our experiment, there was a trend towards reduced weight gain in rabbits. This phenomenon was also observed by other author [20], where hemp seed oil in the quail diet significantly affected final body weight, feed conservation ratio, feed intake and body weight gain. All performance parameters in the hemp seed oil group were lower compared to the control group.
Thank you once again for your time to improve the quality of our manuscript.

Reviewer 3 Report (New Reviewer)
Comments and Suggestions for Authors
The paper addresses an important and timely topic. I found the subject matter of the article fascinating and read the manuscript with great interest. The paper aligns well with the scope of the journal. However, I believe that in its current form, it has several shortcomings.
The introduction provides a good overview of the potential benefits of hempseed cake and its use in animal diets. However, it would be beneficial to include more recent references and a broader discussion on the sustainability and economic impact of using agro-industrial co-products in animal feed.
Lines 46-47: I suggest adding as reference 10.1016/j.heliyon.2024.e31345
Line 82: I suggest also include a statement regarding the general reproduction and litter characteristics, I suggest reading and cite: 10.1080/1828051X.2020.1827990
While the objectives are clear, they could be expanded to include more specific hypotheses. For example, stating expected outcomes for growth performance and reproductive health parameters could provide a clearer direction for the study.
The methodology section is detailed, but it would benefit from additional information regarding:
1) the feed analysis used, please refer to the published paper, I endorse citing for feed analysis techniques: 10.1080/1828051X.2021.1884005; 10.1080/1828051X.2021.1916408 and 10.3168/jdsc.2020-0074
2) the statistical methods used for data analysis. Specifically, the type of statistical tests and any software used should be mentioned. The authors should consider including references to support the statistical methods used in the analysis. Some key areas to address include: 10.29261/pakvetj/2020.067 for normality test; 10.1016/j.vas.2024.100363 for ANOVA; 10.3389/fvets.2024.1332207 for tukey.
The ethical considerations are adequately addressed, but it would be useful to know if any animal welfare scoring was done regularly during the experiment to ensure the well-being of the rabbits.
The results are presented clearly; however, the statistical significance of the findings needs to be more prominently highlighted in the text. Tables and figures are useful but could be supplemented with more detailed descriptive statistics (e.g., standard deviations or confidence intervals).
The discussion section provides a good interpretation of the results but could be strengthened by comparing the findings with those of other similar studies. This comparison would help contextualize the results and their implications better.
Additionally, the potential mechanisms by which hempseed cake could affect the observed parameters should be explored in more depth. This could include a discussion on the bioactive compounds in hempseed cake and their known effects on animal physiology.
The conclusion is concise but could be expanded to include practical recommendations for the use of hempseed cake in commercial rabbit farming. Moreover, suggestions for future research directions would be valuable for readers interested in this field.
The references section should be updated to include more recent studies, particularly those published within the last five years. This will help ensure that the paper is grounded in the latest research.
Author Response
Response to Reviewer 3 Comments
Dear rewiever
Thank you for your good and inspirating remarks.
The paper addresses an important and timely topic. I found the subject matter of the article fascinating and read the manuscript with great interest. The paper aligns well with the scope of the journal. However, I believe that in its current form, it has several shortcomings.
Point 1: The introduction provides a good overview of the potential benefits of hempseed cake and its use in animal diets. However, it would be beneficial to include more recent references and a broader discussion on the sustainability and economic impact of using agro-industrial co-products in animal feed.
Response 1: This aspect was added to the text of Introduction and is marked in red.
Point 2: Lines 46-47: I suggest adding as reference 10.1016/j.heliyon.2024.e31345.
Response 2: We are sorry, but we do not see any relation of our sentence with this reference. Could the reviewer explain this more detaily.
Point 3: Line 82: I suggest also include a statement regarding the general reproduction and litter characteristics, I suggest reading and cite: 10.1080/1828051X.2020.1827990.
Response 3: These reference has now been added.
Pollesel, M., Tassinari, M., Frabetti, A., Fornasini, D., & Cavallini, D. (2020). Effect of does parity order on litter homogeneity parameters. Italian Journal of Animal Science, 19(1), 1188–1194. https://doi.org/10.1080/1828051X.2020.1827990.
Point 4: While the objectives are clear, they could be expanded to include more specific hypotheses. For example, stating expected outcomes for growth performance and reproductive health parameters could provide a clearer direction for the study.
Response 4: This aspect was added to the text of Introduction lines 101-103.
Point 5: The methodology section is detailed, but it would benefit from additional information regarding:
1) the feed analysis used, please refer to the published paper, I endorse citing for feed analysis techniques: 10.1080/1828051X.2021.1884005; 10.1080/1828051X.2021.1916408 and 10.3168/jdsc.2020-0074.
Response 5.1: The above mentioned references concern to dairy cows. However, we used own protocols for our rabbits. Now we added proper infromation about our analyses for rabbits, lines 142-146.
2) the statistical methods used for data analysis. Specifically, the type of statistical tests and any software used should be mentioned. The authors should consider including references to support the statistical methods used in the analysis. Some key areas to address include: 10.29261/pakvetj/2020.067 for normality test; 10.1016/j.vas.2024.100363 for ANOVA; 10.3389/fvets.2024.1332207 for tukey.
Response 5.2: We thank to the reviewer for suggesting three citations of statistical evaluation of our results. However, we have already mentioned all statistical tests and related software used in our study in the sub-chapter 2.6. (lines 243-249).
Point 6: The ethical considerations are adequately addressed, but it would be useful to know if any animal welfare scoring was done regularly during the experiment to ensure the well-being of the rabbits.
Response 6: As it was already described in the M and M section, all the procedures concerning rabbit keeping were done in accordance with ethical guidlines. We did not perform any special welfare scoring, nevertheless, we monitored the temperature and humidity continuously during the study. The water was provided ad libitum.
Point 7: The discussion section provides a good interpretation of the results but could be strengthened by comparing the findings with those of other similar studies. This comparison would help contextualize the results and their implications better.
Response 7: The more references were added to the discusssion.
Point 8: Additionally, the potential mechanisms by which hempseed cake could affect the observed parameters should be explored in more depth. This could include a discussion on the bioactive compounds in hempseed cake and their known effects on animal physiology.
Response 8: The discussion was expanded to include this issue and is marked in red.
Point 9: The conclusion is concise but could be expanded to include practical recommendations for the use of hempseed cake in commercial rabbit farming. Moreover, suggestions for future research directions would be valuable for readers interested in this field.
Response 9: Since our study has a preliminary character and it may require further investigation involving larger number of animals, we avoid to give any practical recommendations for commercial rabbit farming. However, our preliminary results could be inspirating for those researcher, who are interested in the examination of some agro-industrial co-products in the farm animal feeding.
Point 10: The references section should be updated to include more recent studies, particularly those published within the last five years. This will help ensure that the paper is grounded in the latest research.
Response 10: The references no older than 5 years were added to the article and are marked in red.
Thank you once again for your time to improve the quality of our manuscript.

Reviewer 4 Report (New Reviewer)
Comments and Suggestions for Authors
In the study, the authors aimed to investigate effects of adding 5% and 10% hempseed cake on the body weight, sperm concentration, sperm total molitity, sperm progressive motility, spermatozoa characteristics, and serum haemotology and biochemistry indexes in male rabbits. Lots of data is listed in the research, but the paper sholud be thoroughly modified before being accepted.
1. It should be a original article, not a communication. The type of the paper should be changed.
2. The title “The use of hempseed cake as an agro-industrial co-product in rabbit’s diet” is not precise. It looks like a title of review. It is better to change the title as “The effects of adding hempseed cake on body weight…”
3. In this study, only the body weight was investigted. I wonder whether the feed intake and feed-to-gain were affected by hempseed cake. Why was the P value not given in Table 3. Moreover, the SD values are too big. Why?
4. Delete the brackets of (D0)、(D30)、(D60)、(D90)in all the tables, and put the “n=20” into the notes of the tables.
5. The replicates should be listed in Tables 7 to 15.
6. Glu concentrtaions were significantly differently before administration of the experiment among the groups. Why? Whether did it affects the results of the experiments? Please explain.
7. The P values should be given in the Tables from Tables 3 to 15.
Author Response
Response to Reviewer 4 Comments
Dear rewiever
Thank you for your good and inspirating remarks.
In the study, the authors aimed to investigate effects of adding 5% and 10% hempseed cake on the body weight, sperm concentration, sperm total molitity, sperm progressive motility, spermatozoa characteristics, and serum haemotology and biochemistry indexes in male rabbits. Lots of data is listed in the research, but the paper sholud be thoroughly modified before being accepted.
Point 1: It should be a original article, not a communication. The type of the paper should be changed.
Response 1: Initially we submitted our manuscript as an original article. However, during the reviewing process we changed the type of the article for the short communication according to suggestions of one of reviewers and the Editor.
Point 2: The title “The use of hempseed cake as an agro-industrial co-product in rabbit’s diet” is not precise. It looks like a title of review. It is better to change the title as “The effects of adding hempseed cake on body weight…”
Response 2: We are agree with the reviewer, that our original title is rather of global character. So, we have changed our title for the following: „The effects of adding hempseed cake on sperm traits, body weight, biochemical and haematological parameters of rabbit males“.
Point 3: In this study, only the body weight was investigted. I wonder whether the feed intake and feed-to-gain were affected by hempseed cake. Why was the P value not given in Table 3. Moreover, the SD values are too big. Why?
Response 3: The feed intake was not monitored in our experiments, and the animals were fed ad libitum. The P value has now been added under the Table 3. High SD values were reasoned by high variability of body weight among the animals. This aspect is explained in the discussion lines 480-489.
Point 4: Delete the brackets of (D0)、(D30)、(D60)、(D90)in all the tables, and put the “n=20” into the notes of the tables.
Response 4: These comments have been processed in the text.
Point 5: Glu concentrtaions were significantly different before administration of the experiment among the groups. Why? Whether did it affects the results of the experiments? Please explain.
.
Response 5: This difference was minimal and at the lowest level of significance. What is important from our point of view is that the values in all groups at the beginning of the experiment were around 8 mM/L ± 0.5; and all values were within the reference range. Of course, if the difference were greater, it would be worth considering the exclusion of one of the individuals from the experiment, but in this case we did not consider it necessary.
Point 6: The P values should be given in the Tables from Tables 3 to 15.
Response 6: - P-values have been added under every tables.
Thank you once again for your time to improve the quality of our manuscript.

Reviewer 5 Report (New Reviewer)
Comments and Suggestions for Authors
Dear Authors,
The work presented here makes a significant contribution to understanding the potential of hempseed cake in the diet of male rabbits, particularly in relation to reproductive and certain non-reproductive indices. However, I have several comments and suggestions that, if addressed, could enhance the quality and clarity of the information presented in your paper. Please find my detailed comments in the attached PDF file.

Extensive editing of the English language is required.
Author Response
Response to Reviewer 5 Comments
Dear rewiever
Thank you for your good and inspirating remarks.
The work presented here makes a significant contribution to understanding the potential of hempseed cake in the diet of male rabbits, particularly in relation to reproductive and certain non-reproductive indices. However, I have several comments and suggestions that, if addressed, could enhance the quality and clarity of the information presented in your paper. Please find my detailed comments in the attached PDF file.
All minor corrections and spelling errors have been marked in red in the text.
Point 1: Line 39: At what % of supplementation...
Response 1: …and we can recommend the use of hempseed cake at doses up to 10 % in the nutrition and feeding of rabbits.
Point 1: RDWs - and PDWs - .
Response 2: Abbreviations RDWs - red cell distribution width standard and PDWs - platelet distribution width standard were added to the text in M and M section.
Point 3: Tukey is mainly used for large dataset or large sample size. Reviewer suggest the authors to redo the stats using Bonferroni. Bonferroni is conservative and has more power when the number of comparisons is small, whereas Tukey is more powerful when testing large numbers of means.
Response 3: The data were re-calculated using a Bonferroni test.
Point 4: Please provide weekly and overall. Also provide information related to the FI and FCR, weekly and overall. Statistical data is missing in Tables 3-7, please provide.
Response 4: We give the data on weekly and overal weight gain in the table 3. Neither feed intake not feed conversion ratio were measured, because we provide feeding for rabbits ad libitum. Statistical data were added to the footnotes to Tables 3-7.
Point 5: Tables must be self explanatory, thus define abbreviations in the table footer, consider this comment for all the tables.
Response 5: The abbreviations have been addetd in the table footer.
Point 6: Please provide mean value (remove SD for each mean value), and residual standard deviation (RSD) of the means in a new column, and provide p-value even though its not significant. Consider this comment for all the tables displaying results.
Response 6: We have already all our statistical data processed as average ± SD, so that transformation of the data to RSD is complicated in technical aspect. I apologize, but our statistical programe does not allow display RSD, It displays only Count, Average, Median, Standard deviation, Coeff. of variation,Standard error, Minimum, Maximum, Range, Stnd. Skewness and Stnd. Kurtosis.
Point 7: Maintian 3-digits mean values; eg., 27.66 to 27.7; 245.12 to 245, consider this comment for all the tables.
Response 7: This aspect has been taken into account in all tables.
Point 8: Why significant letter is missing? Even if value is intermediate, please provide here and wherever required in the table/s.
Response 8: The letters were added to the tables.
Point 9: In general, the discussion sometimes draws comparisons with ruminants; however, this study was conducted on monogastric. Therefore, the reviewer suggests that the authors focus more thoroughly on monogastric, reduce the emphasis on explaining results from other studies, and discuss the present study in more detail compared to others, explaining the results in line with or in contrast to the findings of this study.
Response 9: In the discussion, we mostly refer to monogastric animals, like rabbits, sheep, chicken, goat. Only once we mention about cattle (polygastric), just in comparison with other animals.
Thank you once again for your time to improve the quality of our manuscript.

Round 2
Reviewer 3 Report (New Reviewer)
Comments and Suggestions for Authors
the paper improved a lot, i do not have any further consideration
Author Response
Thank you very much for your review one more time.
Reviewer 5 Report (New Reviewer)
Comments and Suggestions for Authors
Accept in present form
Comments on the Quality of English LanguageModerate editing of the English language is required.
Author Response
Thank you very much for your review one more time.
This manuscript is a resubmission of an earlier submission. The following is a list of the peer review reports and author responses from that submission.
Round 1
Reviewer 1 Report
Comments and Suggestions for Authors
Unfortunately, in addition to the small errors, there are also some shortcomings in the manuscript, due to which it cannot be accepted for publication. I list some of them below:
L18 and L31: Instead of weight gains per week and the total average weight gains, just write weight gain
L21 and L36: male rabbit
L26: The number of rabbits in total or per group is an important information.
L27-28: Instead of groups E1 and E2, it would be more informative to write E5 and E10 in the entire manuscript.
L80 and L87: This is a bit repetitive, so I recommend writing "housed in individual cages" and "kept in standard metal cages" in one sentence.
L84-86: The number of rabbits per group (n = 10) is too small to examine weight gain. Therefore, these results should be omitted from the article (L217-228).
In the case of semen quantity and quality, it is not possible to know how many samples were included in the statistical evaluation. Semen samples were taken weekly, and those that were not suitable were discarded. If more semen samples from one animal are included in an average, this must be taken into account in the statistical evaluation. If there are only 10 samples in a group, it is unacceptably few. The sample number determines whether the results related to the sperm parameters can remain in the manuscript or not (L229-263). It is definitely noticeable; it happens that the average of one group is twice as big as the other, but the difference is not significant.
L355-471: When evaluating the results (Effect of hempseed cake on weight gain and semen quality), the authors listed a lot of literature data/results in a very long, concise paragraph. In several cases, they described results achieved on animal species (dairy cow, dairy goat, lactating ewes, laying hen) and traits (e.g. milk production, meat quality) that cannot in any way be compared with male rabbits, especially the semen parameters. In other cases, results related to a feed supplement (e.g., flaxseed, rosemary leaves, turmeric) are described, which cannot be compared with hempseed cake, even if whole hemp seed or oil was added to the feed. In all cases, it is a matter of listing the literary results and not of explaining and justifying the results.
For physiological parameters, however, 10 samples/group may be enough. I have not dealt with this part in the review.

Author Response
Response to Reviewer 1 Comments
Dear rewiever
Thank you for your good and inspirating remarks.
Unfortunately, in addition to the small errors, there are also some shortcomings in the manuscript, due to which it cannot be accepted for publication. I list some of them below:
Point 1: L18 and L31: Instead of weight gains per week and the total average weight gains, just write weight gain
Response 1: We accepted the reviewers' comment and the sentence was rewritten.
Point 2: L21 and L36: male rabbit
Response 2: We accepted the reviewers' comment and the sentence was rewritten.
Point 3: L26: The number of rabbits in total or per group is an important information.
Response 3: We accepted the reviewers' comment and the data has been added.
Point 4: L27-28: Instead of groups E1 and E2, it would be more informative to write E5 and E10 in the entire manuscript.
Response 4: We accepted the reviewers' comment and the data was changed.
Point 5: L80 and L87: This is a bit repetitive, so I recommend writing "housed in individual cages" and "kept in standard metal cages" in one sentence.
Response 5: We accepted the reviewers' comment and the sentences were corrected.
Point 6: L84-86: The number of rabbits per group (n = 10) is too small to examine weight gain. Therefore, these results should be omitted from the article (L217-228).
Response 6: We are agree with the reviewer that the number in the group is too small. However, these results are substantial and key for our study, since we monitored effect of hempseed cake along with sperm quality, blood biochemistry and haematology also on live weight of animals. Moreover, substantial part of Discussion is built on the confrontation of our results in relation to the weight gain of rabbits. Therefore, deletion of this part of results could decrease the whole value of our study. Nevertheless, if the reviewer strongly insists on the removal of this results, we would agree with this.
Point 7: In the case of semen quantity and quality, it is not possible to know how many samples were included in the statistical evaluation. Semen samples were taken weekly, and those that were not suitable were discarded. If more semen samples from one animal are included in an average, this must be taken into account in the statistical evaluation. If there are only 10 samples in a group, it is unacceptably few. The sample number determines whether the results related to the sperm parameters can remain in the manuscript or not (L229-263). It is definitely noticeable; it happens that the average of one group is twice as big as the other, but the difference is not significant.
Response 7: We are agree with the reviewer, that we presented unclear number of samples in the M and M section. Prior to experiment we gathered more males than 10 in each group and we monitored them during one month in order to teach them for use an artificial vagina. During this time we eliminated those males which showed worsened sperm quality or were unsuitable in any way. However, to the beginning of our experiments we used in fact only males of good sperm quality, i.e. 10 males per each group. Moreover, we collected samples from each male twice a week, and, therefore, the actual number of samples from each group was 20. We added it now into revised version of the manuscript.
Point 8: L355-471: When evaluating the results (Effect of hempseed cake on weight gain and semen quality), the authors listed a lot of literature data/results in a very long, concise paragraph. In several cases, they described results achieved on animal species (dairy cow, dairy goat, lactating ewes, laying hen) and traits (e.g. milk production, meat quality) that cannot in any way be compared with male rabbits, especially the semen parameters. In other cases, results related to a feed supplement (e.g., flaxseed, rosemary leaves, turmeric) are described, which cannot be compared with hempseed cake, even if whole hemp seed or oil was added to the feed. In all cases, it is a matter of listing the literary results and not of explaining and justifying the results.
Response 8: We just wanted to compare effects of hempseed cake in different animal species. However, we are agree with the reviewer, and we substantially shortened this part of discussion. We made this part of discussion on a more concise and compact form where we just mentioned shortly about effect of hempseed in different farm animal species and we removed part about chicken studies, as it is far from the mammalian object. In part of discussion are described additives, which have similar active substances as hemp, so they could have a similar effect on rabbit sperm quality as these substances, since the effect of hemp on sperm quality in rabbits has not been published so far.
Point 9: For physiological parameters, however, 10 samples/group may be enough. I have not dealt with this part in the review.
Response 9: We agree with this statement.
Thank you once again for your time to improve the quality of our manuscript.

Reviewer 2 Report
Comments and Suggestions for Authors
1. Introduction
The importance of rabbit production worldwide should be further developed, using for example the FAOSTAT references.
The objective of the work must be indicated and if there are other similar studies. References on the physiology of nutrition in rabbits and the anatomy of their digestive system. The working hypothesis is the importance it could have in rabbits from a nutritional, health and even economic point of view.
Include references to sperm production in this species and its influences
2.2 animals
Indicate the dimensions of the cages. It must be indicated whether the commercial food had antibiotics or not and whether those that were added with hemp cake had antibiotics or not. If so, its presence must be justified. The total fiber and protein composition of the commercial food must be indicated. The quantity and quality of the water is not indicated.

Author Response
Response to Reviewer 2 Comments
Dear rewiever
Thank you for your good and inspirating remarks.
Point 1: Introduction
The importance of rabbit production worldwide should be further developed, using for example the FAOSTAT references.
The objective of the work must be indicated and if there are other similar studies. References on the physiology of nutrition in rabbits and the anatomy of their digestive system. The working hypothesis is the importance it could have in rabbits from a nutritional, health and even economic point of view.
Include references to sperm production in this species and its influences.
Response 1: We accepted the reviewers' comment and the objective of the work was added. Also references on the physiology of nutrition in rabbits, the anatomy of their digestive system and references to sperm production in rabbit were append.
Davies Rees R, Davies Rees JAE. Rabbit gastrointestinal physiology. Vet Clin Exot Anim 2003; 6:139–153.
Foote RH, Carney EW. The rabbit as a model for reproductive and developmental toxicity studies
Reprod Toxicol 2000;14:477-493.
Alvariño JMR. Reproductive performance of male rabbits. In: Blasco, A. (Ed.), 7th World Rabbit Congress. World Rabbit Sci 2000;8A:13–35.
Mocé E, Vicente JS, Lavara R, Viudes-de-Castro MP, López M, Bolet G. Characteristics of fresh semen from eight rabbit breeds. In: Vázquez JM. (Ed.), 9th Annual Conference ESDAR. Reprod Domest Anim 2005;40(388):191.
Castellini C, Cardinali R, Dal Bosco A, Minelli A, Camici O. Lipid composition of the main fractions of rabbit semen. Theriogenology 2006;65:703-712.
Point 1: Animals
Indicate the dimensions of the cages. It must be indicated whether the commercial food had antibiotics or not and whether those that were added with hemp cake had antibiotics or not. If so, its presence must be justified. The total fiber and protein composition of the commercial food must be indicated. The quantity and quality of the water is not indicated.
Response 2: The dimensions of the cages were added to the text. The commercial food, equally experimental foods were administered without of antibiotics addition, it was implemented to the text. The total fibre and protein composition of the commercial food is indicated in Table 2. To section M and M was added sentence “The clean tap water was provided using nipple drinkers”.
Thank you once again for your time to improve the quality of our manuscript.

Reviewer 3 Report
Comments and Suggestions for Authors
THis manuscript deals with Hemp seed used as a supplement and its effect on blood parameters and reproducitve capacity of male rabbits. It is an interesting manuscript overall, with a lot of data supporting the conclusion. However, the are some concerns about the manuscript that will need to be adressed before it is ready for publication IMHO.
Let's be specific
L60-62: affirmation needs a reference
L91-93THe sentence is unclear. the table 1 clarifies, but the sentences lead the reader to think that the diets were diluted by the hemseedcake added at 0%, 5% or 10%. Diluting the diets would dilute the nutrients. Lookin at table 1, it does not seem to be the case. The authors need to rephrase and clarify.
table 1: Why EG2 did not have the same proportion of vit-min premix added, as seen in the other treatments?
table2: Since EG2 did have a lower level of vitamins and trace minerals added to it (table 1), it is unlikely that these nutrients provided were the same as the other two treatments. Please clarify that
L175-178: Replace date of blood samplings by the number of days of the the duration of the experiment. For example: D0 (day zero) or Time zero (T0); D30 (day 30) or T30, and so on.
L220: add "numerical" between "slight" and "decrease"
L256-258 : this sentence goes int the discussion section along with table 7
L268-272 + table 9: The reader see that the 3 rabbits initial experimental groups were not equivalent regarding initial blood glucose level. This needs to be adressed in the discussion, not in the results section. Further the blood glucose level was lower after 2 months of trial for EG1 and EG2 as compared to control treatment (table 13). THus, could it be possible that the initial difference at D0 for blood glucose level between treatments is related to the difference between treatments after 2 months? This concern needs to be discussed by the authors.
Discussion section 4.2: The results presented in this manuscript show significant difference for some blood parameters between treatments and with time of blood sampling. However, these differences are not discussed in the manuscript. Therefore, why some blood parameters vary sometimes in opposite directions with time and between treatments? The authors need to adress these findings. Each reasult reported in the tables needs to be discussed.
Comments on the Quality of English LanguageThe language quality is not a concern as far as I am concerned
Author Response
Response to Reviewer 3 Comments
Dear rewiever
Thank you for your good and inspirating remarks.
This manuscript deals with Hemp seed used as a supplement and its effect on blood parameters and reproductive capacity of male rabbits. It is an interesting manuscript overall, with a lot of data supporting the conclusion. However, the are some concerns about the manuscript that will need to be addressed before it is ready for publication IMHO.
Let's be specific
Point 1: L60-62: affirmation needs a reference
Response 1: This affirmation belongs to reference “Sorrentino G. Introduction to emerging industrial applications of cannabis (Cannabis sativa L.). Rend Lincei Sci Fis Nat 2021;32(2):233–243.”
Point 2: L91-93THe sentence is unclear. the table 1 clarifies, but the sentences lead the reader to think that the diets were diluted by the hemseedcake added at 0%, 5% or 10%. Diluting the diets would dilute the nutrients. Lookin at table 1, it does not seem to be the case. The authors need to rephrase and clarify.
Response 2: Lines 91-93 were rephrased. “The rabbits in the 1st experimental group (E5; n=10) were fed granulated mixture including 5% hempseed cake and in the 2nd experimental group (E10; n=10) were fed granulated mixture including 10% hempseed cake, equally without of antibiotics addition”.
Point 3: table 1: Why EG2 did not have the same proportion of vit-min premix added, as seen in the other treatments?
Response 3: Unfortunately, the data was incorrectly copied into the table, the value should be the same as in the C an E1 (E5) group. The data was corrected in the table.
Point 4: table2: Since EG2 did have a lower level of vitamins and trace minerals added to it (table 1), it is unlikely that these nutrients provided were the same as the other two treatments. Please clarify that
Response 4: The question is essentially answered by answer number 2.
Point 5: L175-178: Replace date of blood samplings by the number of days of the the duration of the experiment. For example: D0 (day zero) or Time zero (T0); D30 (day 30) or T30, and so on.
Response 5: We accepted the reviewers' comment and the data were rewritten.
Point 6: L220: add "numerical" between "slight" and "decrease"
Response 6: We accepted the reviewers' comment. The changes are marked in text.
Point 7: L256-258: this sentence goes int the discussion section along with table 7. This comment was implemented to the discussion.
Response 7: We accepted the reviewers' comment and the sentence has been transferred to the discussion.
Point 8: L268-272 + table 9: The reader see that the 3 rabbits initial experimental groups were not equivalent regarding initial blood glucose level. This needs to be adressed in the discussion, not in the results section. Further the blood glucose level was lower after 2 months of trial for EG1 and EG2 as compared to control treatment (table 13). THus, could it be possible that the initial difference at D0 for blood glucose level between treatments is related to the difference between treatments after 2 months? This concern needs to be discussed by the authors.
Response 8: Thank you for your comment, we edited the discussion. Changes are marked in the text.
Point 9: Discussion section 4.2: The results presented in this manuscript show significant difference for some blood parameters between treatments and with time of blood sampling. However, these differences are not discussed in the manuscript. Therefore, why some blood parameters vary sometimes in opposite directions with time and between treatments? The authors need to adress these findings. Each reasult reported in the tables needs to be discussed.
Response 9: Thank you for the comment, of course the individual parameters differed slightly over time, it is about the physiological, health, nutritional status of the organism, and several other influences. Basically, the goal was not to monitor the changes of these parameters over time, because of the aforementioned, we had a control group in the experiment during the entire experiment. That is, we evaluated the experiment after a certain time by comparing it with the control group, which in our opinion is the best and most accurate way of evaluation for the given number of animals. If we were to incorporate statistical evaluations, e.g. between the control group (or experimental group 1) by month, it would introduce more chaos and ambiguity into the results and explanations. We think that if a control group is also included in the samples during the entire experiment, the most significant and authoritative comparison in one sample is only between these groups.
There were added 3 new references:
Rakha, A., Rasheed, H., Altemimi, A. B., Tul-Muntaha, S., Fatima, I., Butt, M. S., ... & Aadil, R. M. (2024). Tapping the Nutraceutical Potential of Industrial Hemp against Arthritis and Diabetes-A Comprehensive Review. Food Bioscience, 104195. https://doi.org/10.1016/j.fbio.2024.104195
Munteanu, C., Mihai, M., Dulf, F., Ona, A., Muntean, L., Ranga, F., ... & Papuc, I. (2023). Biochemical changes induced by the administration of Cannabis sativa seeds in diabetic Wistar rats. Nutrients, 15(13), 2944. https://doi.org/10.3390/nu15132944
Cai, L., Wu, S., Jia, C., Cui, C., & Sun-Waterhouse, D. (2023). Active peptides with hypoglycemic effect obtained from hemp (Cannabis sativa L) protein through identification, molecular docking, and virtual screening. Food Chemistry, 429, 136912. https://doi.org/10.1016/j.foodchem.2023.136912
Thank you once again for your time to improve the quality of our manuscript.
